# Biosynthesis and Industrial Production of Androsteroids

**DOI:** 10.3390/plants9091144

**Published:** 2020-09-03

**Authors:** Rituraj Batth, Clément Nicolle, Ilenuta Simina Cuciurean, Henrik Toft Simonsen

**Affiliations:** Department of Biotechnology and Biomedicine, Technical University of Denmark, 2800 Kongens, Denmark; ribatth@dtu.dk (R.B.); clement.nicolle@etu.unistra.fr (C.N.); cuciurean.1741553@studenti.uniroma1.it (I.S.C.)

**Keywords:** steroid hormones, androsteroids, testosterone, pregnenolone, progesterone, cholesterol biosynthesis, phytosterols

## Abstract

Steroids are a group of organic compounds that include sex hormones, adrenal cortical hormones, sterols, and phytosterols. In mammals, steroid biosynthesis starts from cholesterol via multiple steps to the final steroid and occurs in the gonads, adrenal glands, and placenta. This highly regulated pathway involves several cytochrome P450, as well as different dehydrogenases and reductases. Steroids in mammals have also been associated with drug production. Steroid pharmaceuticals such as testosterone and progesterone represent the second largest category of marketed medical products. There heterologous production through microbial transformation of phytosterols has gained interest in the last couple of decades. Phytosterols being the plants sterols serve as inexpensive substrates for the production of steroid derivatives. Various genes and biochemical pathways involved in phytosterol degradation have been identified in many Rhodococcus and Mycobacterium species. Apart from an early investigation in mammals, presence of steroids such as androsteroids and progesterone has also been demonstrated in plants. Their main role is linked with growth, development, and reproduction. Even though plants share some chemical features with mammals, the biosynthesis is different, with the first C22 hydroxylation as an example. This is performed by CYP11A1 in mammals and CYP90B1 in plants. Moreover, the entire plant steroid biosynthesis is not fully elucidated. Knowing this pathway could provide new processes for the industrial biotechnological production of steroid hormones in plants.

## 1. Introduction

The group of hormones called androsteroids belong to the class of chemical compounds known as steroid hormones. Steroid hormones are produced in mammalian female and male gonads and adrenal glands [1]. They are derived from cholesterol and are biosynthesized in the endoplasmic reticulum (ER) and mitochondria [2]. Androsteroids are lipophilic and bind to transport proteins that increase their half-life and ensure ubiquitous distribution throughout the organisms [2].

Mammalian steroid hormones are classified based on their biosynthesis in respective organs, and sex hormones are mainly produced by ovaries and testes [3]. Androsteroids, the male sex hormones, are mainly produced in testes, ovaries, adrenal glands and placenta [4]. Testosterone is the major androsteroid, and men produce about 7 mg per day in the testis and around 100 µg in the adrenal glands [5]. Females also produce testosterone, but not to the same extent [6].

## 2. Biosynthesis of Steroid Hormones in Mammals

Steroid hormones are biosynthesized from cholesterol via a multiple-step process occurring in the testes and adrenal glands [7] (Figure 1). Alongside other important androsteroids, testosterone appears as the major final product of this highly regulated biosynthesis.

The number of enzymes involved in biosynthesis of steroids is significantly lower than the number of chemical reactions [7] and is catalyzed by two groups of enzymes: cytochromes P450 and dehydrogenases, acting as either aldo-keto reductases or short-chain dehydrogenase/reductases. The interest into steroid hormones in humans has mainly focused on their modes of action, however, after 30 years the biosynthesis of steroid hormones is now almost fully elucidated, with all enzymes and products having been identified [8] (Figure 1).

The first reaction (Figure 1) is the conversion of cholesterol to pregnenolone by the cholesterol side-chain cleavage enzyme CYP11A1. Then, the 17α-hydroxylase activity of CYP17A1 converts pregnenolone to 17OH-pregnenolone. The 17,20 lyase activity of the same enzyme converts this former derivative to dehydroepiandrosterone. Dehydroepiandrosterone is converted to androstenedione by 3β-hydroxysteroid dehydrogenase type 2 (HSD3B2) and followed by the conversion of androstenedione to testosterone by 17β-hydroxysteroid dehydrogenase type 3 (HSD17B3). This reaction of reduction has also been shown to be catalyzed by 17β-hydroxysteroid dehydrogenase type5 (AKR1C3) [9,10]. The resulting testosterone is excreted into the circulation and taken up by genital skin, in which 5α-reductase type 2 (SRD5A2) converts it to dihydrotestosterone (Figure 1).

Testosterone biosynthesis primarily occurs via the conversion of dehydroepiandrosterone to androstenedione and not through the conversion of 17OH-progesterone. It has been shown that the kinetics of human CYP17A1 favors the conversion of 17OH-pregnenolone to dehydroepiandrosterone over that of 17OH-progesterone to androstenedione. The apparent *K_m_* and *v_max_* of CYP17A1 for 17OH-progesterone are 10-fold higher (*K_m_*) and 10-fold lower (*v_max_*) than those for 17OH-pregnenolone [11]. Thus, the catalytic efficiency (*k_cat_/K_m_*) for the 17,20-lyase reaction by one mole of CYP71A1 is about 100-fold higher for 17OH-pregnenolone than for 17OH-progesterone. These results show that strategies for heterologous production of testosterone or steroid derivatives using progesterone as a starting material, instead of cholesterol or other sterols, is likely to be unsuccessful.

The aforementioned pathway is called the “classic pathway” since a second pathway called “the backdoor pathway” was recently described by Miller and Auchus in 2019 [8]. The main differences between these two pathways are the bypass of the usual intermediate steroids dehydroepiandrosterone, androstenedione and testosterone (Figure 1).

Even though steroids are primarily associated with mammals due to early investigations and drug productions, it has been demonstrated that steroids are also natural components of all other eukaryotes including plants, where they also are involved in regulation of the physiological processes related to growth, development, and reproduction [12,13].

## 3. Heterologous Production of Androsteroids

Steroid pharmaceuticals are ranked among the most marketed medical products and represent the second largest category next to antibiotics [14]. The worldwide production of steroid drugs has passed 1 mega ton per year and the global market is around US $10 billion [15].

Androstenedione and 1,4-androstadiene-3,17dione are the most valuable 17-keto steroids, and are used as substrates in chemical synthesis to produce several steroid derivatives, such as testosterone, estradiol, progesterone, cortisone, cortisol, prednisone, and prednisolone [16]. Therefore, a cost-effective production of these products on a large scale is an interesting area of industrial biotechnology.

## 4. Microbial Biotransformation for Production of Steroids

Synthetic chemical methods and biochemical methods starting from natural raw materials containing sterols have already been well described and industrially developed [17]. However, the search for cheaper substrates is continuously being pursued. Phytosterols have gained an increasing importance as raw materials for the synthesis of steroidal drugs. The utilization of cholesterol and phytosterol as the sole carbon source by *Mycobacterium* sp. for growth and proliferation have led to the development of microbial biotransformation processes for production of the majority of steroidal drugs [18].

Phytosterols are inexpensive mixtures of plant sterols mainly of soya origin or produced from tall oil, with sitosterol, stigmasterol, campesterol, and brassicosterol as the major plant substrates [18]. Their abundance in industrial waste and their ability to be substrates for the production of steroidal derivatives through degradation of their aliphatic side chains has opened a new area of research in white biotechnology [19]. Genes and biochemical pathways involved in phytosterol degradation have been identified in many *Rhodococcus* and *Mycobacterium* species [20,21,22]. 

Many phytosterol biotransformation methods have been established: subsections of plants on aqueous medium, two-phase systems, cloud point systems, immobilized biocatalyst systems, and microemulsion and liposome systems [17,19,23]. Nevertheless, market demand still pushes researchers to find more efficient and economical processes by focusing on three major issues: (i) improvement of microbial strain, (ii) utilization of economical substrates, and (iii) process optimization [17].

## 5. Recent Developments in Microbial Biotransformation

Through the development of state-of-the-art bioinformatic analysis and new metabolic engineering approaches, many bacterial strains have been genetically engineered to optimize the metabolism of cholesterol and phytosterol into a steroidal intermediate. The main bioprocesses implemented to produce androsteroids and C-22 steroids using recombinant DNA strategies have been reviewed by Fernández-Cabezón et al. [23]. They described the recent cost-effective strategies to obtain steroid compounds from phytosterols (Table 1).

Manipulation of key enzyme genes involved in the oxidation pathways of phytosterols could improve the production of 17-keto steroid intermediate. Wei et al. [24] identified the main gene encoding 3-ketosteroid-Δ1-dehydrogenase (KstD) in *Mycobacterium neoaurum* NwIB-01 and overexpressed it to increase 1,4-androstadiene-3,17dione purity (Figure 2). The molar yield of 1,4-androstadiene-3,17dione decreased by 15.7%. Engineering of enzymes included in the aliphatic side-chain degradation pathways such as C27-monooxygenase CYP125-3 (Figure 2) has shown to increase androstenedione and 1,4-androstadiene-3,17dione yields in *Mycobacterium neoaurum* [26].

Phytosterol biotransformation with *M. neoaurum* has been improved by manipulation of its methylcitrate cycle, a mechanism of detoxification of intracellular propionyl-CoA, a concomitant toxic byproduct [30]. Propionyl-CoA is the major by-product during β-oxidation of the phytosterol side chains and is toxic in high concentrations [31]. Metabolism of one mole of β-sitosterol yields four moles of FADH2, two moles of Acetyl-CoA, and two moles of propionyl CoA [32,33]. Thus, engineering propionyl-CoA metabolism in mycobacteria through co-expression of the PCC subunit beta and type II NADH dehydrogenase (NDH-II) yielded an economical phytosterol biotransformation [27].

GlnR is a key regulatory protein in the propionyl-CoA pathway in *Mycobacterium*. Genetically manipulating the transcription factors to enhance the methylcitrate cycle [34] again provided an economical androstenedione production strategy. Thus, by addressing the toxic stress and lower production of propionyl, higher yields have been accomplished. 

The maintenance of the redox balance by the NADH/NAD^+^ levels within engineered mycobacteria intracellular environments is considered one of the rate-limiting factors in the phytosterols conversion process. Overexpression of the NADH:flavin oxidoreductase in mycobacteria, and a NADH oxidase from *Lactobacillus brevis*, has been shown as a useful strategy for industrial androstenedione production [32]. Similarly, the heterologous expression of *Bacillus subtilis* NADH oxidase in *M. neoaurum*, along with overexpression of the catalase gene, led to 80% higher yield of 1,4-androstadiene-3,17dione [29]. These studies on toxic stresses and redox balance provide promising strategies for enhanced and robust production of 1,4-androstadiene-3,17dione and androstenedione and other steroid intermediates by the conversion of plant waste materials.

Zhou [35] went further in this direction by combining both previously cited strategies with the development of a state-of-the-art fermentation engineering technique, thereby tackling the three major issues stated by Malaviya and Gomes [17]. A repeated batch fermentation in an expensive surfactant-waste cooking oil-water media was established in 0.4% Tween-80–14% WCO-Water media Systems, leading to a shortened biotransformation period and an improved androstenedione yield.

Engineered strains able to produce testosterone by biotransformation of phytosterols were recently established in Mycobacteria [25]. A *Mycobacterium smegmatis* with stably overexpressed HSD3B from *Comamonas testosteroni* and from *Cochliobolus lunatus* was used (Figure 2), however significant amounts of testosterone were not yet obtained. Lately, Guevara et al. [18] engineered a *Rhodococcus ruber* strain as a cell factory for testosterone production via bioconversion of androstenedione*. Rhodococcus* strains efficiently degrade androstenedione and testosterone, requiring the knockout of enzymes involved in the steroid catabolism (KtsD and 3-ketosteroid 9α-hydroxylase (Ksh)). Furthermore, *R. ruber* exhibits several steroid pathways with unknown steps. Therefore, it is not yet possible to use *R. ruber* as a cell factory to produce testosterone directly from phytosterols.

## 6. Biosynthesis of Androsteroids in Plants

Phytosterols can be precursors for heterologous biosynthesis of steroid hormones [13] such as phytoecdysteroids, brassinosteroids [36], progesterone, testosterone, and its derivatives [13]. Androsteroid functions are not limited to reproduction; they are also involved in controlling plant vegetative development. In addition, the presence of Androsteroid is organ- and species-dependent and the amount is modified during plant developmental steps [37]. Furthermore, extensive studies of testosterone and its derivative occurrence performed by Simons and Grinwich in 1989 [38] in 128 plant species using radioimmunoassay demonstrated their presence in 70% of the plant species. In addition, androsteroids were detected in the seeds of all 128 species [13]. The testosterone biosynthetic pathway in plants occurs in the cytosol, starting from cholesterol. Cholesterol is transformed to pregnenolone after the cleavage of its side chain. Pregnenolone is then converted in androstenedione and finally in testosterone [13]. However, the knowledge on the metabolic pathway of plant steroid hormones, the enzymes involved, and their function is still very partial and much more has to be discovered.

## 7. Biosynthesis of Pregnenolone and Progesterone in Plants

In plants, it has been seen that the growth of *Arabidopsis thaliana* seedlings is enhanced by low concentration of progesterone and inhibited by increasing concentration [20]. In addition, in *Nicotiana tabacum* it was observed that progesterone, testosterone, and estradiol influenced tube elongation and pollen germination [39]. More recently, a precise method was developed using ultra performance liquid chromatography tandem mass spectrometry (UPLC-MS/MS) that detected the presence of all the androsteroids in *Digitalis purpurea* L., *Nicotiana tabacum* L., and *Inula helenium* L. [40]. Thus, the compounds are there. Testosterone, along with epitestosterone and androstenedione, was firstly isolated from pollen of Scotch pine, *Pinus silvestris.* Their presence has been estimated to be around 0.8, 1.1, and 5.9 µg/10 g pollen, respectively [41]. Later, along with progesterone, the presence of these hormones was also shown in *Pinus nigra* using radioimmunoassay and fluorimetry. These results suggest that the amount of testosterone, epitestosterone, and androstenedione oscillates between 0.7 and 0.8 µg/10 g pollen [42]. In elecampane (*Inula helenium* L.) and tobacco (*Nicotiana tabacum*), androstenedione has been detected, respectively, at the concentration of 11 pmol g^−1^ FW and t 7.7 pmol g^−1^ FW [40]. Presence of testosterone was also detected in other pine species such as *Pinus tabulaeformis* and *Pinus bungeana,* along with its occurrence in reproductive organs of *Brassica campestris*, *Juglans regia*, *Ginkgo biloba,* and *Lillium davidii* [43]. The levels of testosterone were highest in *Lilium davidii* at 244 ng/g^−1^ dry weight and lowest in *Pinus bungeana* 11 ng/g^−1^ dry weight. Presence of testosterone in these plants might be associated with development of male gametophyte and pollen germination [43]. The amount of testosterone in each species is summarized in Table 2.

The biosynthesis of pregnenolone was identified in *Haplopappus heterophylius* in 1966 [44], which can be converted to progesterone. Thus, it is likely that the biosynthesis in plants and mammals are similar. This is supported by the presence of the Δ^5^-3β-Hydroxysteroid dehydrogenase/Δ^5^-Δ^4^-ketosteroid isomerase in *Digitalis lanata* [45]. Δ^5^-3β-Hydroxysteroid dehydrogenase converts pregnenolone to isoprogesterone, while Δ^5^-Δ^4^-ketosteroid isomerase converts by isomerization isoprogesterone to progesterone (Figure 3).

In vertebrates, the conversion of cholesterol to pregnenolone is mediated by CYP11A1 homologues [46] (Figure 4).

CYP11A1 perform a triple reaction, including Adrenodoxin reductase, Adrenodoxine, and side-chain cleavage. In plants, the conversion of cholesterol to pregnenolone is not fully elucidated, and at least one enzymatic step is still unknown, even though homologous proteins of Adrenodoxin reductase and Adrenodoxine cleavage have been found in *Arabidopsis thaliana* [47]. CYP11A1 performs the 22R-C hydroxylation of cholesterol, followed by 20R-C hydroxylation (Figure 4B). Both reactions are stereospecific. The last step of the process is the oxidative cleavage of the single bond between C20 and C22, giving pregnenolone (Figure 4B) [48].

CYP11A1 displays a cavity, called a sterol-binding pocket, which interacts with the 3β-hydroxyl group of cholesterol. Cholesterol is bonded in the active site in order to allow the interaction of C22 and C20 with the heme group, and the cholesterol side chain appears bended at C22 and C20 positions. This conformation leads to consecutive hydroxylations, arranging the carbons 22 and 20 above the Heme iron, respectively, at 4.3 Å and 4.5 Å [49].

In plants, the first C22 hydroxylation is performed by CYP90B1, which catalyzes the C-22 hydroxylation of many sterols [50], having a higher affinity to cholesterol (Figure 4B) than for campesterol (Figure 4C) and sitosterol (Figure 4D) [51]. How 22-OH cholesterol gets converted to pregnenolone in plants is still unknown. A major difference between CYP11A1 and CYP90B1 is the stereo specificity. The latter inserts the hydroxyl group in the S position, whereas CYP11A1 is in the R position. However, when cholesterol is the substrate of CYP90B1, R22-Hydroxycholesterol can also be formed (Figure 4B) [52]. In CYP90B1, the active site is characterized by the conformations “side-chain in” and “steroid-core out,” similar to CYP11A1, and for CYP90B1 the 3-hydroxy group hydrogen is bonded with H385 and the carbonyl oxygen of V216 via a water molecule. However, due to the perpendicular position of the CYP90B1 steroid core, the C-22 is located closer to the heme iron (4.1 Å) and gets hydrolyzed, whereas 20-C is too far from the heme group (4.9 Å) to get hydrolyzed, unlike in CYP11A1 where both positions are hydroxylated [52].

The remaining steps of pregnenolone biosynthesis and beyond in plants is still unknown (Figure 3). A full elucidation will benefit the in planta production and possibly establish novel insights into the industrial biotechnological production of steroid hormones. Further, the co-expression analysis of CYP90B1 with other enzymes can also help in the elucidation of the complete pathway in plants. 

## Figures and Tables

**Figure 1 plants-09-01144-f001:**
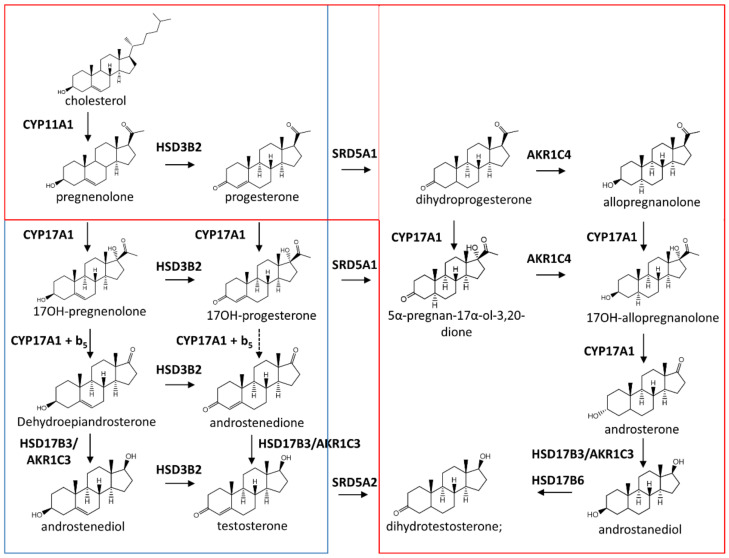
Overview of the biosynthesis of steroids in mammals. This overview combines the classic pathway (in blue) and the backdoor pathway (in red). In the classic pathway, cholesterol is converted into pregnenolone through cleavage of the side chain, followed by multiple steps towards testosterone synthesis. 3β-hydroxysteroid dehydrogenase type 2 (HSD3B2) converts Δ5 steroids to Δ4 steroids. In humans, production of testosterone is favored through 17OH-pregnenolone, as the catalytic efficiency of CYP17A1 is largely higher with this derivative than with 17OH-progesterone. The backdoor pathway bypasses production of dehydroepiandrosterone and androstenedione. Actions of 5α-reductase of type 1 (SRD5A1) and 3α-hydroxysteroid dehydrogenase (AKR1C4) lead to production of androsterone, the major androsteroid of this pathway.

**Figure 2 plants-09-01144-f002:**
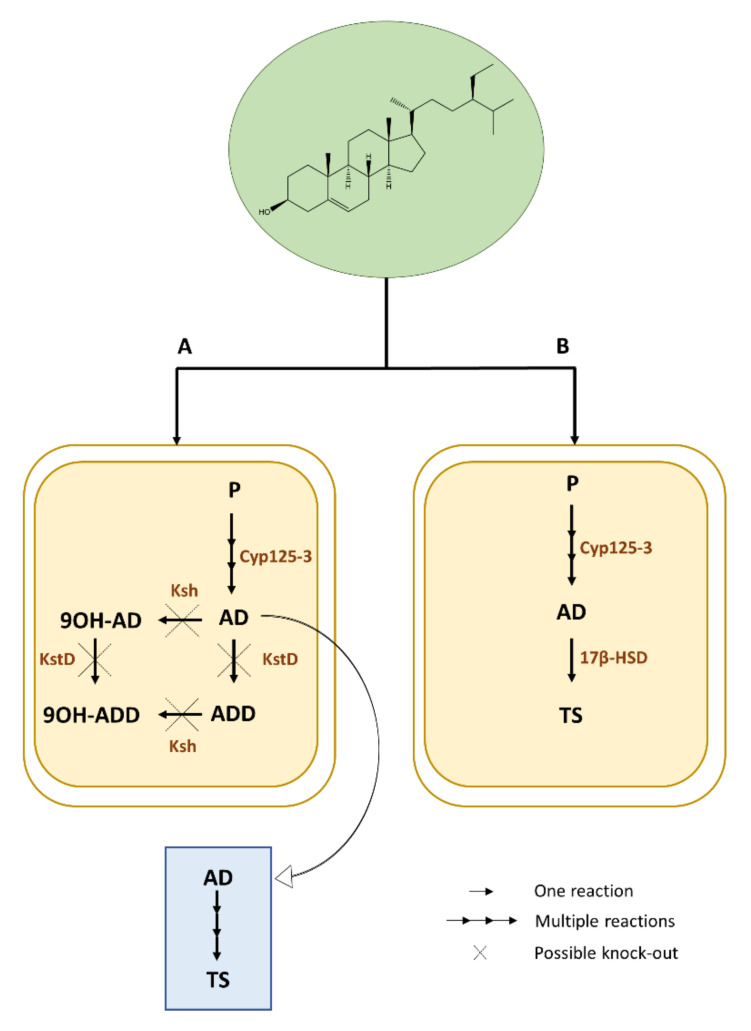
Production of steroid compounds such as testosterone (TS) via a biotransformation process. Phytosterols (P—in green) from agricultural waste, represented here by β-Sitosterol, are used as relevant and cost-effective substrates. (A) They are degraded by microbial cell factories (in yellow), such as mycobacteria, through a multiple-step process where a chemical synthesis (in blue) of TS follows the biotransformation of androsterone (AD), (B) or through a one-step process where microbial strains are engineered to directly convert phytosterols to TS. 9OH-AD, 9α-hydroxy-4-androstene-3,17-dione; 9OH-ADD, 9 alpha-hydroxyandrosta-1,4-diene-3,17-dione; KstD, 3-ketosteroid Δ1-dehydrogenase; Ksh, 3-cetosteroide-9α hidroxilasa; 17β-HSD, 17-β hydroxy steroid dehydrogenase; Cyp125-3, steroid C27-monooxygenase.

**Figure 3 plants-09-01144-f003:**
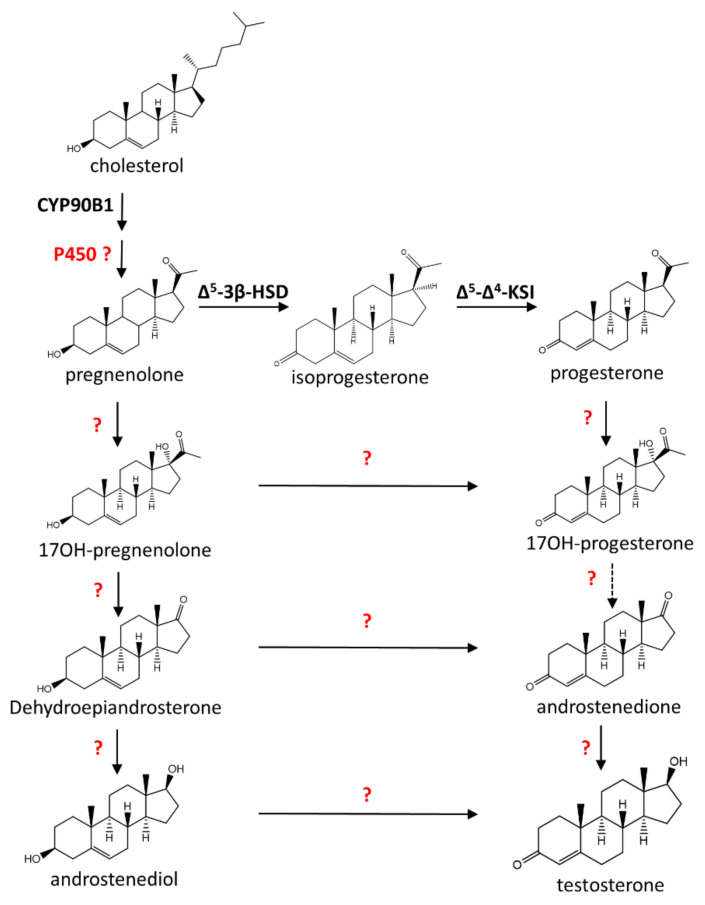
Steroid biosynthesis pathway in plants. This overview shows the interpretation of the classic steroid biosynthesis pathway in plants. The enzyme is responsible for the conversion of cholesterol to pregnenolone and pregnenolone to progesterone, having been characterized in different plant species. However, the full elucidation of the remaining steps is unknown. Δ^5^-3β-HSD, Δ^5^-3β-Hydroxysteroid dehydrogenase; Δ^5^-Δ^4^-KSI, Δ^5^-Δ^4^-ketosteroid isomerase.

**Figure 4 plants-09-01144-f004:**
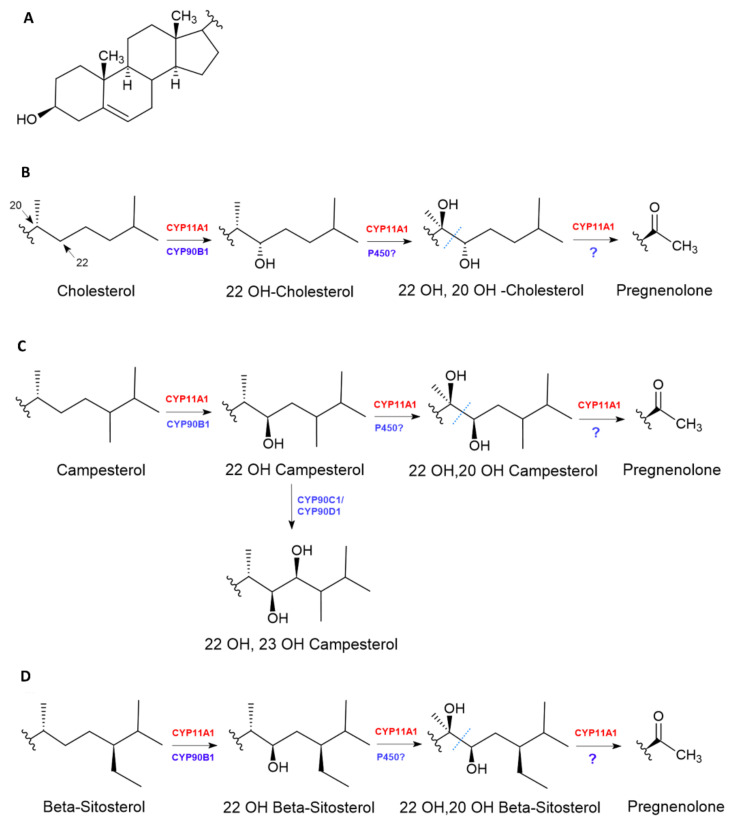
Conversion of phytosterols in pregnenolone. The figure shows the steroid core (**A**) and the possible hydroxylation reactions for phytosterols: cholesterol (**B**), campesterol (**C**) and beta-sitosterol (**D**). In vertebrates the conversion is carried out by CYP11A1 (in red), but it also functions with plants sterols. In plants, the enzymes involved are shown in blue. It is still unknown how the side-chain cleavage between C20 and C22 (represented by dotted blue line) occurs in the plant kingdom.

**Table 1 plants-09-01144-t001:** Current optimized processes to produce steroid intermediates through phytosterols biotransformation. Several examples of strategies implemented to produce steroid intermediates are shown. Main end products obtained at the bioconversions are indicated. ADD, 1,4-androstadiene-3,17-dione; AD, 4-androstene-3,17-dione; 9OH-AD, 9α-hydroxy-4androstene-3,17-dione; 4-HBC, 22-hydroxy-23,24-bisnorchol-4-ene-3-one; TS, Testosterone.

Substrate	Product	Organism	Process	Reference
Soybean phytosterols (stigmasterol, campesterol and β-sitosterol)	AD, ADD	*Mycobacterium neoaurum* NwIB-01	KstD overexpression to increase production and purity of ADD	[24]
Androstenedione or cholesterol	TS	*Mycobacterium smegmatis* mc2155	*Comamonas testosteroni* or *Cochliobolus lunatus* 17β-HSD heterologous overexpression	[25]
Phytosterols mixture: 51.7% of β-sitosterol, 27.2% of stigmasterol, 17.1% of campesterol, and 4.0% of brassicasterol	AD, ADD	*Mycobacterium neoaurum* TCCC 11978	CYP125-3 overexpression	[26]
Phytosterols from untreated cane molasses	AD, 9OH-AD	*Mycobacterium neoaurum* TCCC 11978; *Mycobacterium fortuitum* TCCC 111744	Co-expression of the PCC subunit beta and type II NADH dehydrogenase	[27]
Phytosterols	9OH-AD, 4-HBC	*Mycobacterium neoaurum* ATCC 25795	Neutralization of extra production of ROS through catalase overexpression and mycothiol or ergothioneine augmentation	[28]
Phytosterols	ADD	*Mycobacterium neoaurum* JC-12	Increase NAD+ formation through heterologous expression of B. subtilis nicotinamide adenine dinucleotide oxidase and overexpression of catalase	[29]
Phytosterols	AD	*Mycobacterium neoaurum* TCCC 11978; *Mycobacterium fortuitum* TCCC 111744	Type II NADH dehydrogenase overexpression, repeated batch fermentations	[27]
Androstenedione	TS	*Rhodococcus ruber* Chol-4	*Cochliobolus lunatus* 17-ketosteroid reductase heterologous overexpression	[18]
Phytosterols	AD	*Mycobacterium neoaurum* TCCC 11978	Repression of propionyl-CoA metabolism. 2-methylcitrate cycle pathway prpDBC overexpression and nitrogen transcription regulator GlnR deletion	[30]

**Table 2 plants-09-01144-t002:** Testosterone and its natural occurrence in different plant species.

Steroid	Amount	Origin	Reference
Testosterone	0.08 µg/g^−1^	*Pinus silvestris*	[41]
	0.07–0.08 08 µg/g^−1^	*Pinus nigra*	[42]
	11 ng/g^−1^	*Pinus bungeana*	[43]
	244 ng/g^−1^	*Lilium davidii*	[43]

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
