# Peer review of "Biosynthesis and Industrial Production of Androsteroids"

_plants, 2020, doi:10.3390/plants9091144_

Round 1
Reviewer 1 Report
A well-written article was summarizing current knowledge in the field of science. However, I would welcome some suggestions on how research in this area should develop.
Author Response
Thank you for your nice comments. The research in this area can further develop by the co-expression analysis of CYP90B1 with other enzymes, which will help in elucidation of complete pathway in plants. We have also mentioned this in the revised manuscript.
Reviewer 2 Report
The authors deal in detail with the biosynthesis of mammalian steroids and its differences in animals and plants, only a minor part is devoted to biotechnological production. Therefore, I recommend changing the title of the article to better reflect its essence. They also focus not only on Androsteroids but also on Progestagens.
Minor comments:
I recommend improving the quality of the figures (schemes).
Table 1: recommend moving it to chapter 5.
Legend of Fig.2: the description of branch B is missing.
Author Response
Thanks for the nice words on our review. We have now updated the title to something more appropriate “Biosynthesis and Industrial Production of Androsteroids”.
As for the minor comments:
The improved quality figures have now been added to the manuscript
As per your suggestion, we have now moved the Table 1 to chapter 5.
Legend of Fig.2: the description of branch B is missing. Thank you for bringing this to our notice, Figure 2 now has the proper description of Branch 2.
Reviewer 3 Report
This review article handles the production and possible pathways of steroids in animals and plants as well as microbial conversions of these steroids. It is very informative descriptions on some pathways involved in production of androsteroids between animals and plants. Overall, it was well organized and written. I would like to recommend authors to add brassinosteroid to this manuscript because it is comparatively well-studied in plants comparing to other steroids in plants.
Author Response
Thanks for the nice comments. However, the brassinosteriods are biosynthesized in a very different manner and possibly with a very special regulation, thus not added to the schemes, and it is already well reviewed.